# Advances, Perspectives and Potential Engineering Strategies of Light-Gated Phosphodiesterases for Optogenetic Applications

**DOI:** 10.3390/ijms21207544

**Published:** 2020-10-13

**Authors:** Yuehui Tian, Shang Yang, Shiqiang Gao

**Affiliations:** Department of Neurophysiology, Physiological Institute, University of Wuerzburg, 97070 Wuerzburg, Germany; yuehui.tian@uni-wuerzburg.de (Y.T.); shang.yang@uni-wuerzburg.de (S.Y.)

**Keywords:** cyclic nucleotides, phosphodiesterases (PDEs), optogenetics, cAMP, cGMP

## Abstract

The second messengers, cyclic adenosine 3′-5′-monophosphate (cAMP) and cyclic guanosine 3′-5′-monophosphate (cGMP), play important roles in many animal cells by regulating intracellular signaling pathways and modulating cell physiology. Environmental cues like temperature, light, and chemical compounds can stimulate cell surface receptors and trigger the generation of second messengers and the following regulations. The spread of cAMP and cGMP is further shaped by cyclic nucleotide phosphodiesterases (PDEs) for orchestration of intracellular microdomain signaling. However, localized intracellular cAMP and cGMP signaling requires further investigation. Optogenetic manipulation of cAMP and cGMP offers new opportunities for spatio-temporally precise study of their signaling mechanism. Light-gated nucleotide cyclases are well developed and applied for cAMP/cGMP manipulation. Recently discovered rhodopsin phosphodiesterase genes from protists established a new and direct biological connection between light and PDEs. Light-regulated PDEs are under development, and of demand to complete the toolkit for cAMP/cGMP manipulation. In this review, we summarize the state of the art, pros and cons of artificial and natural light-regulated PDEs, and discuss potential new strategies of developing light-gated PDEs for optogenetic manipulation.

## 1. Introduction to Cyclic Nucleotides and Signal Transduction

In mammalian cells, first messengers such as light, nitric oxide (NO), and hormones can regulate intracellular second messenger cyclic adenosine 3′-5′-monophosphate(cAMP) and cyclic guanosine 3′-5′-monophosphate (cGMP) levels, which further affect the vision signaling, muscle contraction, cardiovascular, memory, and many other functions [1,2,3,4]. cAMP is produced by adenylyl cyclases (ACs), which are divided into six classes [5,6]. Class III adenylyl cyclases are found in animals and some bacteria, and they are the most popular ACs due to their important roles in human health [7]. Most members (AC1-9) of class III ACs are integral membrane proteins and involved in signal transduction, while AC10 (or sAC) is the only soluble class III AC [6]. AC10 can be activated by bicarbonate and plays an important role in sperm motility and fertilization capacity [8,9]. Some Class III ACs’ activities are regulated by intracellular Ca^2+^ concentrations through protein kinases C (PKC) and calmodulin (CaM) [10,11,12]. cGMP-producing guanylyl cyclases (GCs) also comprise membrane-bound (type 1) and soluble (type 2) forms [13]. Single membrane-spanning GCs, named particulate GCs (pGCs), exist in animals and participate in sensory processing [14,15]. For example, pGC-G was proved to be a thermosensory protein [16]. Soluble guanylyl cyclase (sGC) is activated by nitric oxide (NO) and participates in physiological processes such as vasodilatation [17,18,19].

The cAMP or cGMP can regulate multiple cellular physiological processes (Figure 1). Notably, cNMP (cAMP and cGMP) can participate in several regulatory processes by activating downstream components like protein kinase A or G (PKA or PKG) and cyclic nucleotide-gated (CNG) channels [20]. cAMP can also modify the Exchange protein activated by cAMP (Epac) [21] and the Popeye domain containing (Popdc) family to regulate cellular events [22,23,24]. For example, the catalytic subunit of PKA can be released after the binding of cAMP to the regulatory domain and phosphorylate downstream target proteins, e.g., the phosphorylase kinase (PhK) to regulate glycogenolysis [25]. PKA can also be transported into the nucleus to activate cAMP-response element binding protein (CREB) and initiate gene expression [26,27]. As one of the intracellular cGMP targets, PKG is highly expressed in different tissues or cell types, such as smooth muscle, cardiomyocytes, platelets, endothelial, and neuronal cells [28]. PKG induces smooth muscle relaxation in response to NO signaling and lowers blood pressure [29]. PKG stimulation is one of the potential therapeutic strategies for cardioprotection [30].

Early in the 1980s, scientists had already found that cAMP and PKA function in subcellular compartments [31]. PKAs are tethered by different A-kinase anchoring proteins (AKAPs) to distinct cell compartments, where different subcellular signalosomes are organized [32,33,34]. Such signalosomes are identified in different subcellular microdomains and assembled by different proteins [35,36,37]. Recent developed cAMP and PKA sensors helped to observe the compartmented cAMP signaling in more detail [38,39,40,41,42,43]. Sensors for cGMP were developed behind cAMP sensors because of the lower cGMP abundance in most cell types. However, several cGMP sensors are now well developed for monitoring intracellular cGMP [44,45,46].

In subcellular microdomains, cAMP or cGMP can be hydrolyzed rapidly due to robust localized phosphodiesterases (PDE) activities [47]. As crucial components of specific signalosomes, PDEs modulate the diffusion and the spatial gradients of cNMP, ensuring the compartmented signaling. In addition, the hydrolysis activities of PDEs can be further regulated by binding of cNMP (e.g., PDE2, 5, 6, 10 and 11) [48], Ca^2+^/CaM (e.g., PDE1), or phosphorylation by PKA or PKG (e.g., PDE1, 3 and 4) [49,50,51]. To study the rapid processes that regulate cGMP or cAMP levels in subcellular compartments, spatiotemporally precise methods are needed. It was expected that light-induced PDEs could be designed and applied in addition to light-gated nucleotide cyclases, to regulate cGMP or cAMP levels and dissect the related physiological processes.

## 2. Therapeutic Targeting of PDEs

In mammalian cells, 11 primary PDE members (PDE1–11) were identified as a superfamily, with over 100 different isoforms due to multiple transcription start sites and alternative splicing. Some of them hydrolyze both cAMP and cGMP (including PDE1–3, 10, 11), while others are cAMP (PDE4, 7, 8) or cGMP (PDE5A, 6, 9A) specific [49]. They share relatively conserved *C*-terminal catalytic PDE domains but vary much in *N*-terminal regulatory modules. This has effects on individual signaling roles, such as intracellular localization and cell- or tissue-specific expression [52].

Dysregulation of PDEs often leads to defects in compartment-specific cNMP signaling and causes diseases [53]. More importantly, no two PDE isoforms shares the exact same substrate specificity and expression/localization profile. The diversity and uniqueness make PDEs attractive targets for therapeutic intervention. One therapeutic strategy for some diseases related to PDEs is searching for inhibitors which directly reduce substrate binding affinity in the *C*-terminal PDE catalytic domains. For instance, PDE5 inhibitors perform the vasodilatory functions and might provide possibilities for recent clinical trials in some diseases, such as pulmonary hypertension and chronic heart failure [53,54,55].

However, increasing evidence suggests that effective therapies might require selective and precise manipulation of local PDE activities [53]. The other therapeutic direction would be to uncover and leverage the regulating mechanisms of the variable *N*-terminal regions. The *N*-terminal modules of certain mammalian PDEs have different effects, such as heterologous protein–protein interactions, the interactions within PDEs, and cyclic nucleotide binding [56,57]. For instance, PDE2, 5, 6, 10, and 11 have *N*-terminal GAF domains. The cyclic nucleotides binding to the GAF domains can change the overall conformation and regulate the PDE activities [48]. The phosphorylation in the *N*-terminal regulatory regions of PDE3 and PDE4 can dramatically impact their activity and cooperation with other proteins in signalosomes [49]. Notably, some diseases are accompanied by abnormal cAMP increase. For example, based on in vitro kidney cell models, the autosomal dominant polycystic kidney disease (ADPKD) with elevated cAMP levels promotes cyst formation and leads to renal failure. One allosteric activator compound of PDE4 has the potential to lower the cAMP levels and limit the cAMP-mediated signaling pathways. This inhibits the cyst formation after tests in human embryonic kidney (HEK) 293 and Madin Darby Canine Kidney (MDCK, ATCC) cells [58]. Manipulating the *N*-terminal regulatory modules or upstream interventions could give new insights into the regulating mechanism of PDE catalytic activity.

## 3. Light-Regulated PDEs

The inhibitors, activators, and modulators of PDEs have been successfully applied in both basic researches and clinical trials. However, chemical drugs may encounter non-selective and crosstalk problems, especially when targeting the relative conserved catalytic domains. Due to the free diffusion effect, it is challenging to restrict the chemicals to specific subcellular regions or to certain cells in a tissue. In addition, chemical approaches also exhibit poor temporal resolution and lack reversibility. On the contrary, optogenetic manipulation of cNMP provides reversible control with unprecedented spatiotemporal precision. Light-gated production of cNMP has been well established. However, light-regulated PDEs are under development to complete the toolkit for optogenetic cNMP manipulation.

### 3.1. Indirect Light Regulation of PDE Activity in Visual Phototransduction

In the retina of vertebrate eyes, rod and cone cells respond to different light wavelengths and intensities and transfer visual information to neural signals. Type II vertebrate rhodopsins, as a member of G protein (guanine nucleotide-binding protein)-coupled receptor (GPCR) superfamily A, play key roles in regulating visual systems in the disc membrane of rod and cone cells. Photon absorption by rhodopsin initiates the visual signaling cascade (Figure 2). The activated rhodopsin binds to G protein, causing it to dissociate from the bound guanosine diphosphate (GDP) and bind guanosine triphosphate (GTP). The GTP-bound Gα subunit dissociates from Gβγ subunits and becomes active [59]. PDE6 is anchored in the photoreceptor outer segment membranes. It will be activated by the GTP-bound Gα protein, thus decrease the cGMP concentration and down-regulate CNG channel activities in the plasma membrane. The following hyperpolarization of the membrane potential in photoreceptor cells enables neurotransmitters to release to different cells in downstream and trigger neuronal signaling in the brain [60,61,62]. In the dark, the inactive form of rhodopsin leaves the Gα protein in its GDP-bound form, which inhibits the PDE6 and reduces its cGMP hydrolysis activity. Furthermore, GC is activated by guanylate cyclase-activating proteins (GCAP) at low concentrations of Ca^2+^. This will restore cGMP levels in the cytoplasm and reopen CNG channels in the dark.

Among the 11 PDE family members in mammals, only PDE6 is indirectly regulated by light through G-protein coupled receptor in the disc membrane of rod and cone cells [60,61,62], and no direct light regulation has been reported in animal PDE superfamily. Due to the complicated signaling cascade from the light sensing to PDE activity regulation, the GPCR-G protein-PDE pathway cannot be used as a universal tool for light manipulation of cGMP in different cells.

### 3.2. Artificial Light-Activated PDE (LAPD)

The signal transduction from the sensor domain to the effector domain shares a similar conformation-changing mechanism across different signaling receptors. Therefore, replacing the *N*-terminal chemosensor domain of PDE by a photosensor module may endow the photosensitivity. Based on this strategy, LAPD was firstly engineered by combining a bacteria photosensor module with the catalytic domain of human PDE2A [63] (Figure 3a). Crystallography and biochemical study revealed that the linker region between GAF-A, B and catalytic domain plays a key role in regulating the activity of PDE2A. The binding of cGMP to GAF-B induces significant movement of the coiled-coil linker between GAF-B and catalytic domain and activates PDE2A allosterically [64].

Phytochromes can be regulated by red and far-red light that can penetrate deeper into tissues. Red light and far red could reach 4–5 mm beneath the skin surface, while blue light can barely penetrate 1 mm into the skin tissue [66]. The bacterial phytochromes (BPhys) use biliverdin as chromophore, which is a natural product of heme and universally available among cell types. This makes it applicable without exogenous chromophore supplementation. Two stable phytochrome intermediates, absorbing red light (Pr) and far-red light (Pfr), regulate the activity of the effector domain differently. The first trial of fusing *Pa*BPhy light-sensing domain and the *Hs*PDE2A catalytic domain unfortunately fails to yield soluble protein. Substitution of the *Pa*BPhy photosensor module by that from *Deinococcus radiodurans* (*Dr*BPhy) and slight modulation of the coiled-coil linker region generated the first light-regulated PDE: LAPD (Figure 3a). Upon red-light illumination, LAPD exhibits up to four-fold and six-fold catalytic activity increase towards cAMP and cGMP, respectively. Preliminary applications showed that LAPD allowed optical control of cAMP and cGMP levels in CHO cells and zebrafish embryos [63]. In addition, the red-light elevated catalytic activity of LAPD can be reverted by far-red light irradiance. However, this photoconversion is not complete [63].

To engineer new LAPDs with improved properties, Stabel et al. systematically conducted substitution of either the *N*-terminal photosensor module or the *C*-terminal PDE effector module and modification of the linker regions [67]. A suite of LAPD variants was engineered. Among the variants, *Dr*-*Bt*PDE2A exhibits enhanced reversibility of photoactivation as well as the highest photodynamic range [67]. Expressing LAPD, *Dr*-*Bt*PDE2A and several other variants in HEK cells enable regulating cNMP-dependent physiological processes such as the gating of CNG channels [67].

### 3.3. Direct Light-Gated PDEs (RhoPDEs) from Nature

A rhodopsin-phosphodiesterase gene fusion was found in the genome of a choanoflagellate, *Salpingoeca rosetta* [68]. The protein, named Rh-PDE or RhoPDE (Figure 3b), was expressed in HEK293 cells, and its cGMP and cAMP hydrolysis ability was found to increase 1.4-fold and 1.6-fold respectively with light illumination [69]. The hydrolysis activity of RhoPDE is maximally activated by 492 nm, and it is ~10-fold more active towards cGMP than cAMP. However, a following study using purified proteins suggested that light regulation is absent in RhoPDE [70]. Later it was found that RhoPDE is clearly activated by light with an unusual mechanism: light illumination primarily increases its substrate affinity rather than the maximal turnover [71]. Additionally, we found that the hydrolysis activity for cGMP is ~100 times higher than for cAMP. Both cGMP and cAMP hydrolysis activities can be increased to ~five-fold under light illumination at low substrate concentrations [71]. Different from the classical rhodopsins with seven transmembrane helices (TMs), RhoPDE shows an 8 TMs topology with cytosolic localization of both *N*- and *C*-terminal proved by immunofluorescence microscopy experiments, bimolecular fluorescence complementation (BiFC) experiments [70,71] and a very recent structure study [72].

Brunet et al. found more RhoPDEs from different species of choanoflagellates. Four RhoPDE homologs were discovered from *Choanoeca flexa* sp. nov. In *C. flexa*, they found that the choanoflagellate forms cup-shaped colonies that invert their curvature in response to changing illumination conditions through a rhodopsin-cGMP signaling pathway [65] (Figure 3b). Moreover, from sequenced transcriptomes database, other choanoflagellates species also encode RhoPDEs [73]. Collectively, eight new RhoPDE homologs were identified separately in *C. flexa* (Cf1-4), *Microstomoeca roanoka* (*Mr*1), *Acanthoeca spectabilis* (*As*1), and *C. perplexa* (*Cp*1-2) [65]. All these RhoPDEs are predicted to have a similar 8-TM topology like *Sr*RhoPDE. After expression of these RhoPDEs in HEK293 cell, *Cf*Rh-PDE1, *Cf*Rh-PDE4, and *Mr*Rh-PDE exhibited light-enhanced cGMP hydrolysis activity. However, the activities of *Cf*Rh-PDE1 and *Cf*Rh-PDE4 are not significant. *Mr*Rh-PDE displays similar light activity to *Sr*RhoPDE, but higher dark activity. *As*Rh-PDE, the one lacking the conserved retinal binding lysine residue, shows constant cAMP-specific PDE activity without light regulation [74].

The comparison of currently characterized light-gated PDEs was summarized in Table 1. Generally, all the light-gated PDEs showed considerable high dark activity and the activation ratio is maximally at ~6, which is not ideal for tight optogenetic manipulation.

## 4. Light-Regulated Nucleotide Cyclases

Optogenetic manipulation of cNMP offers reversible control with unprecedented spatiotemporal precision. Light-regulated cAMP/cGMP-producing cyclases exist in a large variety in nature and outperform the conventional chemical cyclase activators in various applications [75,76,77,78,79,80]. The first photoactivated adenylyl cyclase (PAC) was discovered from studying the photophobic responses of *Euglena gracilis*, and proven to be functional in *Drosophila melanogaster* and *Caenorhabditis elegans* cholinergic neurons [75,81]. Later discovered bPAC from the soil bacterium *Beggiatoa* shows much smaller size, lower dark activity, and higher light activity in comparison with *Eu*PAC, making it a versatile tool for manipulating and studying cAMP-mediated processes [76].

In term of light-regulated GCs, *Be*CyclOp (alternative name: *Be*RhGC) is the first identified enzyme rhodopsin consisting of microbial rhodopsin domain and GC domain. *Be*CyclOp is a tightly regulated cyclase with a very high photodynamic range. Co-expression of *Be*CyclOp with CNG channel (TAX-2/4) in *C. elegans* muscle induces body contraction upon light illumination [77]. Its homologue from *Catenaria anguillulae* CaCyclOp exhibits better photostability and higher GTP turnover when expressed in hippocampal neurons compared to *Be*CyclOp [79]. In contrary with light activated GCs, light inhibited GCs called 2c-Cyclops are also identified in green algae *Chlamydomonas reinhardtii* and *Volvo**x carteri*. cGMP synthesis is kept going on in the dark, which requires the phosphoryl transfer from the histidine kinase domain to the response regulator domain. Light illumination disrupts this process and inhibits cGMP production [78]. Interestingly, both light-activated and light-inhibited GC all show a similar 8-TM topology. In addition, light regulated AC and GC can be converted to each other via mutations, further expanding the cyclase toolbox [79,82].

However, light-gated nucleotide cyclases cannot be used to decrease cNMP levels in heterogeneous cells. Thus, light-gated PDEs are needed to complete the toolkit.

## 5. Strategies for Engineering New Light-Regulated PDEs

Current available light-activated PDEs, artificial LAPD and natural RhoPDE, showed similar drawbacks of relative high dark activity and low light to dark activity ratio (Table 1). Further studies of natural light-gated PDEs will deepen the understanding of animal vision evolution and provide potential optogenetic tools for precise cNMP manipulation. Moreover, optogenetic inhibition of endogenous PDEs in addition to pharmaceutical inhibitors will be of great advantage. Here we provide some further possible strategies for engineering of new light-gated PDEs.

### 5.1. Allosteric Light Regulation

The light-oxygen-voltage (LOV) domains are found in various protein sensors, which respond to environmental changes in plants and bacteria. The most widely used and best-studied LOV domain comes from the second LOV domain of *Avena sativa* phototropin 1 (AsLOV2) [83]. It comprises ~125 amino acids with a chromophore-binding pocket for the covalent adduct flavin. Two α helices are flanking in the *N*- and *C*-terminal of *As*LOV2, named A′α and Jα helices, respectively [84,85]. In the dark stage, the *C*-terminal Jα helix is caged in the LOV core motif, while it can be exposed under blue light illumination [86]. This light-triggered conformational change makes the *As*LOV2 domain suitable for designing light controllable protein in an allosteric manner. For example, Wu et al. fused a GTPase Rac1 with the LOV2 domain, obtaining a photoactivatable Rac1 (PA-Rac1). PA-Rac1 is sterically blocked from interacting with its effectors in the dark. Light illumination induces the movement of Jα and subsequently unbinding of Rac1 from LOV2, allowing Rac1 to bind with its effector [87]. Similarly, it is feasible to engineer light-regulated PDEs by combining PDEs and LOV2 with a systematical screening of the proper fusion strategies (Figure 4a). Moreover, in the red/far-red range, linking the allosteric transition of the phytochrome to PDE activity regulation has been proved very effective in LAPDs engineering [63,67].

Light-induced allosteric regulation could also be applied to target endogenous PDEs. Pioneering work by Dagliyan et al. demonstrated that insertion of LOV2 domain to solvent-exposed loops on proteins of interest enables switching the proteins between active and inactive states by light [88]. Inspired by this, Gil et at. engineered a class of opto-nanobodies (OptoNBs) via inserting a modified LOV domain into the selected loop sites of the camelid single domain antibodies (aka nanobodies) with a similar scaffold. The binding of OptoNBs to target proteins can be enhanced or inhibited upon blue light illumination. Expressing these OptoNBs in cells allows reversibly binding to endogenous intracellular targets, modulating signaling pathway activity [89]. Accordingly, inserting the modified LOV variant into proper loop sites of nanobodies against specific PDEs could also generate a suite of light-switchable Opto-αPDEs (anti-PDEs). When the binding site of these Opto-αPDEs locates near to the active site of PDEs, Opto-αPDEs could be potentially applied to inhibit the endogenous PDEs with light application (Figure 4b). Targeting endogenous PDEs would be of great value for the studies of subcellular cAMP/cGMP signaling.

### 5.2. Light-Induced Translocation

Besides allosterically regulation, light-induced changes in protein oligomeric states could also be implemented in engineering light-regulated PDEs. A series of photosensor-derived interaction domains spanning from ultraviolet (UV) to far-red light range provide multiple choices for engineering light-regulated PDEs. In the blue light range, a TULIPs (tunable light-inducible dimerization tags) system was developed based on the *As*LOV2 domain and an engineered PDZ domain (ePDZ). A peptide epitope was fused after the Jα helix, which can then interact with the cognate PDZ domain in a light-dependent manner [90]. An updated iLID (improved light-induced dimer) system was established with less crosstalk to endogenous signaling pathways than TULIPs. A short bacteria SsrA peptide, with only seven residues, was embedded in the *C*-terminal of Jα helix in *As*LOV2 domain. Blue light illumination exposes SsrA from LOV2 domain, allowing it to bind its natural interaction partner SspB. The engineered iLID system shows an over 50-fold increase in the binding affinity after light illumination [91]. Through introducing point mutations in SspB, the binding affinity of iLID system could be further adjusted [92]. Fusing PDE to one component of the iLID and a specific targeting sequence to its binding partner could be able to recruit PDEs at specific subcellular compartment by light illumination (Figure 4c).

Other similar light-induced protein interaction systems can also be applied in a similar way. Cryptochrome derived cryptochrome 2/cryptochrome-interacting basic-helix-loop-helix (Cry2/CIB1) dimerization system offers an alternative approach for blue light-mediated protein interaction. The two components can dimerize under blue light illumination in the sub-second range, while the reversion lasts for minutes [93]. This system can also be activated by two-photon microscopy, enabling in vivo application. In the red spectral window, the binding of phytochrome B (PhyB from *Arabidopsis thaliana*) to its natural interaction partner PIF3 is induced by red-light irradiance and reversed under far-red light exposure or dark state [94]. On the contrary, the binding and dissociation between bacterial phytochrome BphP1 and its partner PpsR2 or Q-PAS1 is stimulated by far-red and red light, respectively [95]. In addition, a palette of available spectrum separated light-induced dimerization systems allows simultaneous control of discrete cell signaling by different light. The red light regulated BphP1/Q-PAS1 and blue light regulated LOV system have been applied to tridirectionally translocate protein between the cytoplasm, nucleus and plasma membrane [95]. Applying BphP1 and LOV derived light-regulated PDEs, it would also be possible to achieve dual-color control of cNMP at specific confinement of the cell.

### 5.3. Light-Gated Recovery of Split PDEs

In addition, Cry2/CIB1 system has been used to reconstitute split protein fragments and recover the activity in a light-dependent manner. After fusion of two split Gal4 fragments with Cry2 and CIB1 and co-expression in yeast, the reporter gene expression can be induced by blue light [93]. The Cry2/CIB1 modules were also able to recover a split Cre recombinase to increase DNA recombination efficiency by light [93]. We already found that split fragments of *Hs*PDE2A catalytic domain could recover the cGMP/cAMP hydrolysis activity upon co-expression (unpublished data). Therefore, the Cry2/CIB1 and other light-induced dimerization systems could be fused with the split *Hs*PDE2A fragments and recover its hydrolysis activity in a light-dependent pattern at desired subcellular localization when combining with specific targeting strategy (Figure 4d). Other light-induced protein–protein interaction systems like iLID and PhyB/PIF3 could be applied in a similar way.

### 5.4. Light-Gated Uncaging of PDEs

Additional to these light-regulated two-component systems, fluorescent protein Dronpa with light-dependent changes in oligomerization state could also be applied to engineer light-regulated PDE. Zhou et al. have shown that fusion of tetrameric Dronpa at both ends of Cdc42 or protease cages their functions in the dark, while light-induced Dronpa dissociation allows uncaging and functioning [96]. Moreover, the improved dimeric variants pdDronpa dimerize in violet light and dissociate in cyan light [97]. Fusing two pdDronpa copies at rationally selected positions in the kinase domain generates photo-switchable kinase [97]. Similarly, attaching two pdDronpa at locations flanking the active site of PDE, a single-chain light-switchable PDE could be engineered. Violet light illumination induces the formation of an intramolecular dimer, thereby caging PDE’s activity, whereas cyan light dissociates the dimer and exposes the active site for hydrolyzing cNMP (Figure 4e). A similar Z-lock system comprising LOV and Zdk domains could also be adapted for generating reversible, light-controlled steric inhibition of the active sites of PDEs (Figure 4f). Previous work has demonstrated that attaching Zdk and LOV2 to the *C* and *N* termini of cofilin respectively could effectively occlude the cofilin active site in dark. Upon irradiation, the dissociation of Zdk and LOV frees the active site [98].

Principally, there are plenty of methods to generate light-gated PDEs for optogenetic applications. However, all these conceptions may require tremendous trials and intensive optimizations to obtain an ideal light-gated PDE for robust applications. Ideal tools can be obtained in a “trial and error” way or by high throughput screening.

## 6. Applications of Light-Gated PDEs

In parallel to engineering light-regulated PDEs, fully exploiting the potential of those perspective optogenetic tools is of equal importance. The enzymatic activity and photodynamic range of each specific light-regulated PDE must be carefully considered at the very beginning of applications. Unlike the light-gated ion channels or pumps with no detectable dark activity, existing light-regulated enzymes often show considerable dark activity. These might lead to changes of cNMP level already in the dark, which is often cell type and expression level-dependent. Developing tightly light-regulated PDEs is the first step to improve this. Real applications with existing light-gated PDEs can be improved by selecting the tools with proper photodynamic range and manipulating the expression level in targeted cells.

Increasing evidence suggests that the cNMP regulated events are precisely controlled in distinct subcellular confinement through recruiting isoform-specific PDEs into the specialized signalosomes [32,99,100,101]. The superior spatiotemporal precision and tunable activity afford researchers the ability to study the cell signaling in unprecedented detail, even in a quantitative manner [102]. For example, when regulating the activity of CNG channel or mimicking the functions of plasma membrane-localized endogenous PDEs, membrane-integral RhoPDEs could be good choices. Employing LAPD enables regulating the cytosol cNMP level. Manipulating cAMP levels by LAPD provides insights into the key roles of cAMP in capacitation of mouse sperm [103]. LAPD can also be combined with bPAC and the cAMP biosensor mlCNBD-FRET to study how the local cAMP contributes to cilia length regulation [104]. Moreover, the light-induced dimerization system allows recruiting PDEs to an intended compartment when one part of the dimer is fused with a specific targeting module (e.g., mitochondria, Endoplasmic Reticulum, or nucleus, etc.). Rost et al. have summarized in detail a number of general principles and specific motif information for subcellular targeting of photosensitive proteins [105]. It should be noted that simple fusion of the targeting motif might not always successfully bring the actuator into expected subcellular localization. Moreover, fusing the targeting motif may also alter the properties of PDEs per se. Accordingly, it is crucial to select a proper linker between the targeting motif and the client protein.

More interestingly, co-application of spectrum separable light-regulated cyclases and light-regulated PDEs would enable bi-directional control of the level of cNMP, either at close or discrete locations. In addition, numerous genetically encoded cAMP and cGMP sensors confer optical visualization of the distribution and dynamics of cAMP and cGMP [106,107]. The combination of spectrum compatible light-gated cyclases, PDEs with fluorescent biosensors [108] holds great promise to simultaneously manipulate and map the dynamics of cNMP signaling in live cells in a superior precise and quantitative manner.

## 7. Conclusions

cAMP and cGMP play essential roles in cell division, differentiation, growth, and death. Cyclic nucleotide PDEs are widely distributed in the animal kingdom, hydrolyzing the ubiquitous second messengers, cGMP and/or cAMP. Therefore, PDE enzymes are crucial to manipulate concentrations of both second messengers to maintain normal responses, thus being regarded as important therapeutic targets. Different regulatory modules in the *N*-termini of most PDEs function diversely to regulate PDE activities through ligand binding, oligomerization and kinase recognition/phosphorylation. Indirect regulation of PDE activities has been well studied in animal vision systems. Direct-coupled light regulation of PDE inside one protein has also been discovered recently in some protists, the ancestor of animals. Further studies into nature rhodopsin PDEs will help to elucidate the vision evolution.

Optogenetic strategies for regulating PDE activities could give new insights to regulate cNMPs accurately in the cellular microdomains. Artificial light-gated PDEs have been developed in a deliberate way in addition to the well-established light-gated nucleotide cyclases. Further developments of superior Opto-PDEs with tighter light regulations are in demand. Furthermore, optogenetic inhibiting or targeting endogenous PDEs will be of great value to basic research, and have therapeutic importance. Many newly developed optogenetic systems can be applied for light manipulation of PDEs. However, intensive efforts are needed to advance this field.

## Figures and Tables

**Figure 1 ijms-21-07544-f001:**
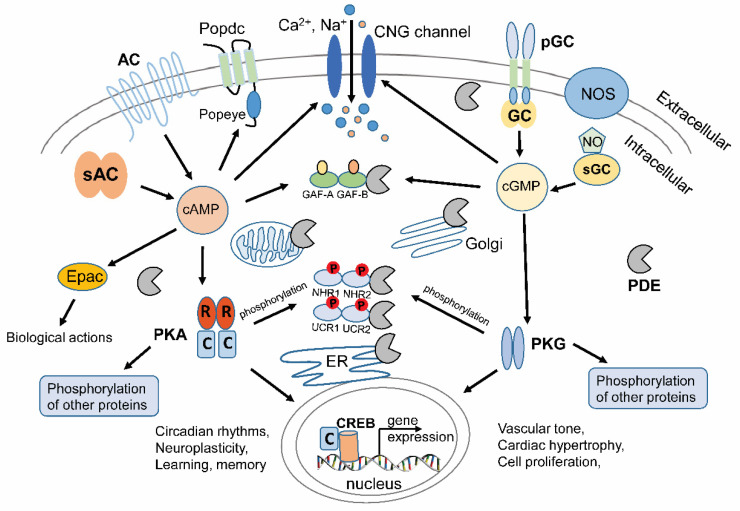
Cyclic nucleotides (cAMP and cGMP) and their typical signal transductions in animal cells. cAMP is produced by membrane-integrated ACs or soluble AC. sGC can be activated by NO produced by NOS and generates cGMP. Membrane-bound GCs respond to external signaling molecules and regulate cGMP production. Main downstream effectors mediated by cAMP and cGMP were depicted. The CNG channels with Ca^2+^/Na^+^ permeability in the plasma membrane can be activated by cAMP or cGMP. As cAMP-specific effectors, Popeye domain containing (Popdc) proteins consist of three transmembrane domains and a cytoplasmic cAMP binding Popeye domain. Popdc plays key roles in maintaining the function of striated muscle cells. In the cytoplasm, various PDEs have GAF domains, GAF-A and GAF-B, in the *N*-terminal, which can bind either cGMP (for PDE2, 5, 6 and 11) or cAMP (for PDE10), thus regulating *C*-terminal catalytical activity. PKA and PKG can further regulate downstream protein phosphorylation and CREB transcription factors for certain gene expression. PKA related gene expressions have effects on circadian rhythms, neuroplasticity, learning, and memory, etc. PKG related gene expressions mainly play roles in the cardiovascular system, such as vascular tone, cardiac hypertrophy and cell proliferation etc. PKA and PKG can phosphorylate the NHR domain of PDE3 or UCRdomain of PDE4 and regulate their catalytic activities. Different PDE isoforms are localized in different microdomains, like ER, mitochondria, Golgi, different cytosolic hotspots, and sub-plasma membrane areas. Note that *N*-terminal modules of different microdomain-targeted PDEs are not depicted in the figure. cAMP: cyclic adenosine 3′-5′-monophosphate; cGMP: cyclic guanosine 3′-5′-monophosphate; AC: adenylyl cyclase; GC: guanylyl cyclase; sGC: soluble guanylyl cyclase; NO: nitric oxide; NOS: NO synthase; CNG: cyclic nucleotide-gated; Popdc: Popeye domain containing; PDE: phosphodiesterase; GAF: cGMP-specific phosphodiesterases/adenylyl cyclases/FhlA; PKA: protein kinase A; PKG: protein kinase G; CREB: cAMP-response element binding protein; NHR: *N*-terminal hydrophobic regions; UCR: upstream conserved region; ER: Endoplasmic Reticulum.

**Figure 2 ijms-21-07544-f002:**
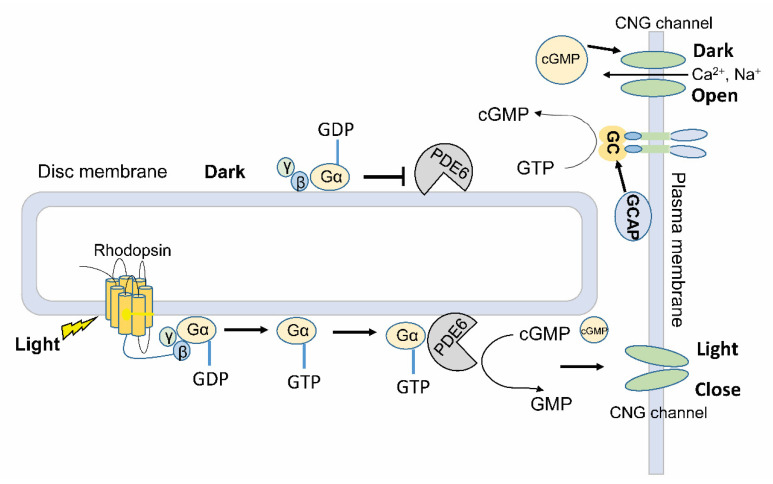
PDE6 is involved in the phototransduction of vertebrate visual system. The disc membrane and plasma membrane depicted here are localized in photoreceptor cells of retina. Key proteins related to phototransduction cascades are shown in the membranes. The photosensor rhodopsin is integrated into the disc membrane, where G proteins (Gα, β, γ) and PDE6 are attached. In the plasma membrane, CNG channels permeable to Ca^2+^ and Na^+^ stay in an open or close state depending on the dark or light conditions. GC and GCAP are shown in the plasma membrane to maintain the cellular cGMP levels. G proteins: guanine nucleotide-binding protein; GDP: guanosine diphosphate; GTP: guanosine triphosphate; GMP: guanosine monophosphate; GCAP: guanylate cyclase-activating proteins.

**Figure 3 ijms-21-07544-f003:**
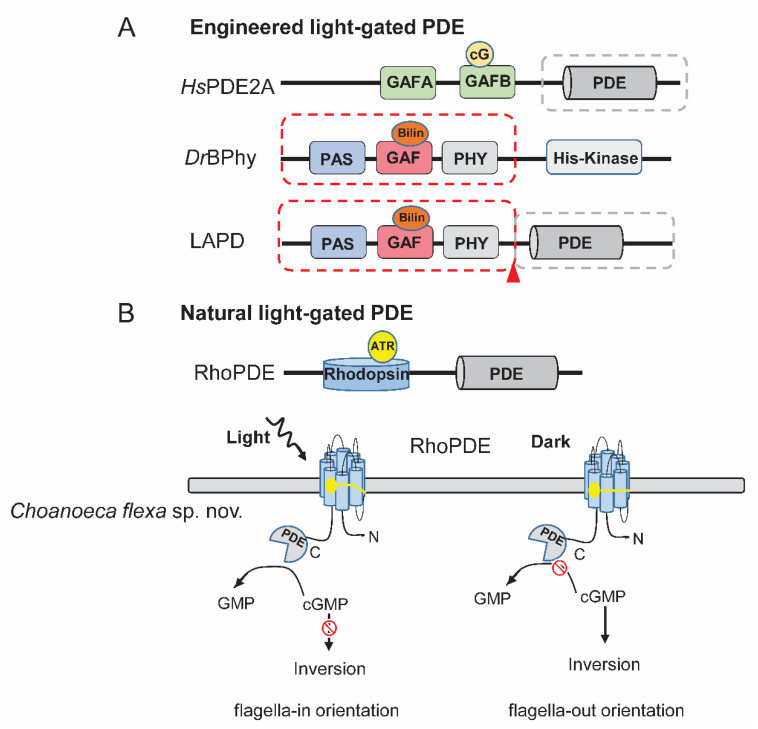
Engineered and natural light-gated PDEs. (**A**) The schematic model of generating the artificial light-gated PDE, LAPD. The LAPD comprises the light sensor modules from *Dr*BPhy (labeled by red dashed box) and PDE catalytic domain from *Hs*PDE2A (labeled by gray dashed box). The encircled cG indicates cGMP. The chromophore is shown with encircled Bilin. The red triangle indicates the connection point. (**B**) The schematic model of natural light-gated PDE. The chromophore is shown with encircled ATR. In a newly discovered protist *Choanoeca flexa* sp. nov, the RhoPDE can be activated by light to hydrolyze cGMP and trigger the inversion to flagella-in orientation. In the dark, cGMP levels will be maintained and flagella-out orientation is kept. The lower cartoon in B is made according to [65]. LAPD: light-activated PDE; Bilin: biliverdin; PAS: Per-Arnt-Sim domain; ATR: all-trans-retinal.

**Figure 4 ijms-21-07544-f004:**
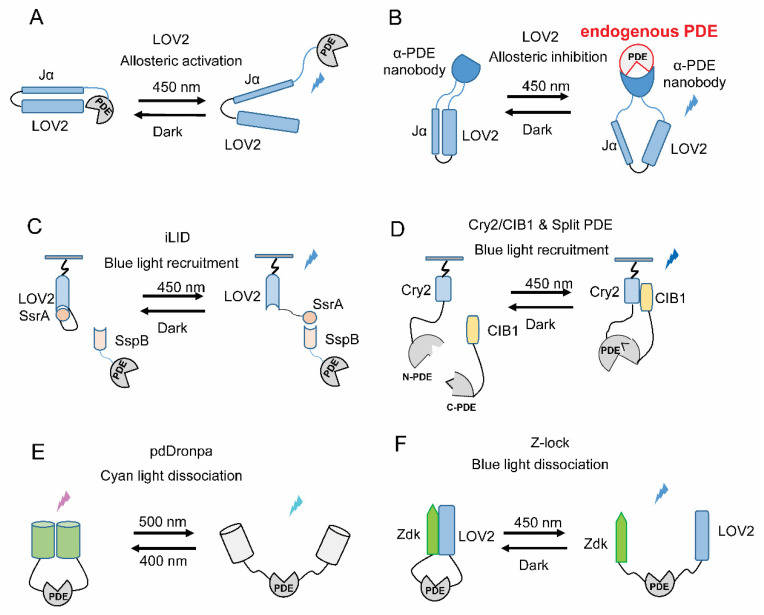
Potential strategies to design new light-regulated PDEs. (**A**) LOV2 based allosteric activation: tight linkage to the LOV domain can block the function of PDE, while the light-induced conformational changes in the LOV domain recover its function. (**B**) LOV2 based allosteric inhibition: LOV2 domain fused at the loop area of anti-PDE single domain antibody (α-PDE nanobody), blue light illumination triggers conformational changes in the nanobody, leading to binding of nanobody to PDE and inhibiting or regulating endogenous PDEs. (**C**) Light-mediated translocation with iLID (LOV2-SsrA/SspB) system. The LOV2-SsrA part can be tethered to membrane fractions in certain organelles. Targeted PDEs could be fused to the corresponding dimeric partner SspB. This could potentially achieve the recruitment of PDEs to a certain subcellular compartment by light illumination. (**D**) Reconstitution of Split PDE fragments. Two split fragments *N*-PDE and *C*-PDE are fused with Cry2 and CIB1, respectively. Blue light can induce the dimerization of Cry2/CIB1 and recover the full PDE hydrolysis activity. (**E**) pdDronpa system: The catalytic domain of PDE are fused with two pdDronpa proteins in *N*- and *C*-terminus or other proper positions. Cyan light-induced dissociation free the active site of PDE while violet light-triggered association block the active site. (**F**) Z-lock system: Zdk and LOV2 are fused at the *N*- and *C*-terminus or other proper positions of PDE, blocking PDE active site in dark. After blue light illumination, Zdk and LOV unlinked, uncaging the active site. LOV: light-oxygen-voltage; iLID: improved light-induced dimer; Cry2: cryptochrome 2; CIB1: cryptochrome-interacting basic-helix-loop-helix 1.

**Table 1 ijms-21-07544-t001:** Comparison of Km, Kcat, and dynamic range of light-gated PDEs.

Light-Gated PDEs	Km	Kcat, Vmax or Turnover	L/D Ratio	References
LAPD	Dark (cGMP): ~440 μM	Dark turnover (cGMP): ~ 42 s^−1^	~6 (cGMP)	[63]
	690 nm (cGMP): ~340 μM	Red 690 nm turnover (cGMP): ~252 s^−1^	~3.6 (cAMP)	
	Dark (cAMP): ~470 μM	Dark turnover (cAMP): ~30 s^−1^	-	
	690 nm (cAMP): ~180 μM	Red 690 nm turnover (cAMP): ~108 s^−1^	-	
*Dr-Bt*PDE2A	-	Red 670 nm s^−1^ turnover (cGMP): ~225 s^−1^	Red/far-red	[67]
		Far-red 780 nm turnover (cGMP): ~38 s^−1^	~6 (cGMP)	
*Sr*RhoPDE	Dark (cGMP): ~80 μM	Dark turnover (cGMP): ~12 s^−1^	2–6 * (cGMP)	[71]
	473 nm (cGMP): ~13 μM	Blue 473 nm turnover (cGMP): ~28 s^−1^	~5 (cAMP)	
*Mr*Rh-PDE	-	Dark (cGMP): ~600 pmol·min^−1 #^	~1.1 (cGMP)	[74]
		520 nm (cGMP): ~672 pmol·min^−1 #^	~1.7 (cAMP)	
		Dark (cAMP): ~145 pmol·min^−1 #^	-	
		520 nm (cAMP): ~250 pmol·min^−1 #^	-	

* The ratio is changing depending on the substrate concentration. # Protein amount is not provided. For the other RhoPDEs, *Cf*Rh-PDE2, *Cf*Rh-PDE3, *Cp*Rh-PDE1, and *Cp*Rh-PDE2 showed no enzymatic activity. *As*Rh-PDE is specific for cAMP hydrolysis but without light-regulation. The activity of *Cf*Rh-PDE1 and *Cf*Rh-PDE4 was not significant [74]. Km: the michaelis constant; Kcat: number of substrate molecules turned into product per enzyme site per minute; Vmax: maximum velocity; L/D: light activity to dark activity.

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
