# Peer review of "Advances, Perspectives and Potential Engineering Strategies of Light-Gated Phosphodiesterases for Optogenetic Applications"

_ijms, 2020, doi:10.3390/ijms21207544_

Round 1

Reviewer 1 Report

The review by Yuehui et al. deals with the interesting and timely topic of optogenetic manipulation of phosphodiesterase (PDEs) activity and provides information that is of interest to a wide group of scientists concerned with cyclic nucleotide signalling. There are however several shortcomings. Below are some suggestions that the authors may want to consider in order to improve the quality of the work.

  • There are multiple grammatical errors and awkward sentences that make the reading rather difficult. The authors should consider seeking assistance for grammar proofreading and style check to ensure that the text is clear and coherent.
  • The focus of the review is largely on application of molecular engineering methodologies that have been applied to other proteins and that, although in principle can be applied to PDEs, mostly have not been carried out in this context or results have not been published. This should be reflected in the title and made clear throughout the text.
  • Paragraphs 1 describes, in quite generic terms, the concept of compatmentalised cNMP signalling. Some additional detail on the organization of signalosomes, role of anchoring proteins and of PDEs in the local hydrolysis of cNMP would be useful to understand the potential of the methodologies described.  
  • Paragraph 2 contains a number of statements that are incorrect. For example, line 80, the authors state that ‘’ heart failure is caused by upregulation of PDE5; line 84, ‘PDE9 inhibitors are selected for the treatment of diabetes or Alzheimer’s disease’; line 94, ‘some diseases are caused by increase in cAMP’ and ADPKD is given as an example. Although some of these links have been suggested and discussed in the literature and there is some experimental evidence in support, they are far from being conclusively demonstrated and generally accepted. The text should be rephrased to more accurately reflect the current knowledge.
  • As the applicability of the different optogenetic approaches to the manipulation of PDE activity is largely speculative at this stage, it would be useful to have a more detailed description of what are the potential hurdles that would need to be overcome for these applications and what are the possible limitations. It would also be useful if the authors could give a sense of pros and cons of different approaches.
  • The biggest challenge of light-gated enzymes is the considerable dark activity: author should provide more detail about how this aspect can be improved and provide more information on current results.
  • Some information on how deep red and far-red light penetrate tissues would be useful
  • Figures: the quality of the figure could be further improved; the resolution of some character is not ideal. Fig 1 could benefit from examples of localised PDE isoforms. Figure legends should include full definition of abbreviations and acronyms used (e.g. define NHR and UCR in figure 1, PAS in figure 3). Line 284, it should be Figure 4C-4F, not 3F.

Author Response

We would like to thank for the fast and fair review process and especially for your insightful and detailed comments and suggestions, which clearly helped to improve our manuscript very much.

Here are our responses to your valuable comments.

The review by Yuehui et al. deals with the interesting and timely topic of optogenetic manipulation of phosphodiesterase (PDEs) activity and provides information that is of interest to a wide group of scientists concerned with cyclic nucleotide signalling. There are however several shortcomings. Below are some suggestions that the authors may want to consider in order to improve the quality of the work.

Comments and Suggestions:

There are multiple grammatical errors and awkward sentences that make the reading rather difficult. The authors should consider seeking assistance for grammar proofreading and style check to ensure that the text is clear and coherent.

ANSWER: We have gone through the text carefully and improved the English. We hope the language is more clear and coherent now.

The focus of the review is largely on application of molecular engineering methodologies that have been applied to other proteins and that, although in principle can be applied to PDEs, mostly have not been carried out in this context or results have not been published. This should be reflected in the title and made clear throughout the text.

ANSWER: We introduced the light-regulation of PDE in animal visual systems, artificial and natural light-gated PDEs in chapter 3. However, two shortcomings of Optogenetic applications are: 1) high dark activity of existing light-activated PDEs and 2) no available method for light-inhibiting endogenous PDEs. Thus, we provided potential strategies for generating new light-gated PDEs in section 5 (previous chapter 4). We have added “Potential Engineering Strategies” in the title to make this part clear. We have also added a new Table 1 (page 7, line 229) of existing light-gated PDEs and a new brief section 4 of “Light-regulated nucleotide cyclases” (line 235-257) to enrich the contents.

Paragraphs 1 describes, in quite generic terms, the concept of compatmentalised cNMP signalling. Some additional detail on the organization of signalosomes, role of anchoring proteins and of PDEs in the local hydrolysis of cNMP would be useful to understand the potential of the methodologies described. 

ANSWER: We have added additional details on the organization of signalosomes, role of anchoring proteins etc. in lines 75 to 82 (clean version) of the revision.

Paragraph 2 contains a number of statements that are incorrect. For example, line 80, the authors state that ‘’ heart failure is caused by upregulation of PDE5; line 84, ‘PDE9 inhibitors are selected for the treatment of diabetes or Alzheimer’s disease’; line 94, ‘some diseases are caused by increase in cAMP’ and ADPKD is given as an example. Although some of these links have been suggested and discussed in the literature and there is some experimental evidence in support, they are far from being conclusively demonstrated and generally accepted. The text should be rephrased to more accurately reflect the current knowledge.

ANSWER: We have rephrased the paragraph 2 (line 99-106) and ADPKD part (lines 115-120 of old version) in paragraph 3 of section 2, removed the improper statements about PDE9 inhibitors (lines 83-86 of old version).

As the applicability of the different optogenetic approaches to the manipulation of PDE activity is largely speculative at this stage, it would be useful to have a more detailed description of what are the potential hurdles that would need to be overcome for these applications and what are the possible limitations. It would also be useful if the authors could give a sense of pros and cons of different approaches. 

ANSWER: We have added comments in lines 367-370. We think that there are no obvious pros and cons of different methods and “Ideal tools can be obtained in a “trial and error” way or by high throughput screening.”

The biggest challenge of light-gated enzymes is the considerable dark activity: author should provide more detail about how this aspect can be improved and provide more information on current results.

ANSWER: All existing light-gated PDEs showed high dark activity, thus a low L/D ratio. To our knowledge and experimental trials, there is no single promising method to reduce the dark activity. As we stated in line 367-370, Ideal tools can be obtained in a “trial and error” way or by high throughput screening.

Some information on how deep red and far-red light penetrate tissues would be useful.

ANSWER: We have added this information in lines 179-180.

Figures: the quality of the figure could be further improved; the resolution of some character is not ideal. Fig 1 could benefit from examples of localised PDE isoforms. Figure legends should include full definition of abbreviations and acronyms used (e.g. define NHR and UCR in figure 1, PAS in figure 3). Line 284, it should be Figure 4C-4F, not 3F.

ANSWER: We have increased the resolution of all figures. We have added subcellular areas like ER, mitochondria and Golgi as possible subcellular PDE hot spots. However, we did not define exactly which PDE situates in which area, because different isoforms of the same PDE family could localize in different subcellular areas and different PDEs can also stay in one hot spot according to the literature. We have provided the full definition of NHR, UCR in Figure 1 and PAS in Figure 3. Figure 4C-4F was corrected.

In the end, we thank you again for all the constructive reviews.

Reviewer 2 Report

In this review, Tian et al. survey the currently available palette of directly light-regulated cAMP/cGMP-specific phosphodiesterases (PDE), the biological role of these cyclic nucleotides, pertinent applications, and future directions for improving the performance of light regulation. Overall, the review is well conceived and structured, but some parts feel a little disjointed, and it is unclear whether all parts add to the paper. In particular chapter 4. on the design strategies suffers from being a tad disorganized and very condensed. Given that all design approaches have been discussed in numerous review articles before, it is not clear whether revisiting the same ground here adds much. However, given the prospective target audience of the article, there may be some benefit, and I hence lean towards retaining this section. That said, it would arguably help to concentrate on general design strategies that may yield light-regulated PDEs, rather than delving into too much detail. For instance, fig. 4 shows in panels C-F essentially very similar approaches; likewise, panels H and I refer to the same principal design idea. Similarly, the main text could be reduced to a description of the principal strategies that one may consider applying. By contrast, I missed a clear statement, maybe a table, as to which properties of the light-regulated PDEs await optimization, e.g., kcat, KM, dynamic range etc. Another aspect that the manuscript could elaborate on in more detail are photoactivated adenylyl (and guanylyl) cyclases which exist in larger variety. A somewhat more detailed description of them (without too many details) appears warranted. Finally, the recently discovered new Rho-PDEs are mentioned but not described in any detail -- a comparison with respect to their properties would greatly add to the article.

Apart from minor comments listed below, the manuscript is written in a balanced, informed, yet easily accessible manner. It will provide a good resource for researchers interested in applying light-regulated PDEs in their research. After suitable revision, the article should hence be ready for publication in the International Journal of Molecular Sciences.

general: English use is good but not perfect. Recurring issues include missing articles, use of singular vs. plural, and often overly long and convoluted sentences.

9: Add ‘the’ before ‘second messengers’.

27-37: This paragraph should introduce the term ‘particulate’ for the transmembrane ACs. (it actually features as an abbreviation in line 34 but is not explained.)

figure 1: Could be expanded by popeye proteins which bind cAMP for regulation.

67: What is meant by ‘in subcellular levels’?

73: ‘Therapeutic regulation’ is a strange term, consider revising.

84-86: Unclear why this inhibitor is described in that much molecular detail when nowhere else in the manuscript this would be the case.

100-108: Not sure whether this paragraph should not be moved to the subsequent chapter.

122: Rephrase ‘... leaves the G_alpha protein in its GDP-bound form, which ...’.

157-159: Detail on original construction of LAPD not needed to this extent, i.e. remove this sentence.

175-178: This part should be moved to the beginning of the section.

211: ‘abundantly exist’ is an exaggeration. As requested above, a brief list of available PAC/PGC options would help.

214: ‘engineering of new’

257: ‘suite’

262: ‘translocation’

339-344: This part would better fit in chapter 4.

325-338: Two recent papers by Wachten (pubmed ids 32579112 and 31252584) seem highly relevant for subcellular localization and issues connected to targeting of optogenetic tools.

Author Response

We would like to thank for the fast and fair review process and especially for your insightful and detailed comments and suggestions, which clearly helped to improve our manuscript very much.

Here are our responses to your valuable comments.

In this review, Tian et al. survey the currently available palette of directly light-regulated cAMP/cGMP-specific phosphodiesterases (PDE), the biological role of these cyclic nucleotides, pertinent applications, and future directions for improving the performance of light regulation. Overall, the review is well conceived and structured, but some parts feel a little disjointed, and it is unclear whether all parts add to the paper. In particular chapter 4. on the design strategies suffers from being a tad disorganized and very condensed. Given that all design approaches have been discussed in numerous review articles before, it is not clear whether revisiting the same ground here adds much. However, given the prospective target audience of the article, there may be some benefit, and I hence lean towards retaining this section. That said, it would arguably help to concentrate on general design strategies that may yield light-regulated PDEs, rather than delving into too much detail. For instance, fig. 4 shows in panels C-F essentially very similar approaches; likewise, panels H and I refer to the same principal design idea.

ANSWER: We have simplified Figure 4 by removing Figure 4D-F, and mentioned them in the text only. We have kept Figure 4H and I (new 4E and F) because 4I is relative new.

Similarly, the main text could be reduced to a description of the principal strategies that one may consider applying. By contrast, I missed a clear statement, maybe a table, as to which properties of the light-regulated PDEs await optimization, e.g., kcat, KM, dynamic range etc. Another aspect that the manuscript could elaborate on in more detail are photoactivated adenylyl (and guanylyl) cyclases which exist in larger variety. A somewhat more detailed description of them (without too many details) appears warranted.

ANSWER: We have added a new Table 1 (page 7, line 229) of existing light-gated PDEs and a new brief section 4 of “Light-regulated nucleotide cyclases” (line 235-257) to enrich the contents.

Finally, the recently discovered new Rho-PDEs are mentioned but not described in any detail -- a comparison with respect to their properties would greatly add to the article.

ANSWER: We have added comparison of new Rho-PDEs in lines 220-224

Apart from minor comments listed below, the manuscript is written in a balanced, informed, yet easily accessible manner. It will provide a good resource for researchers interested in applying light-regulated PDEs in their research. After suitable revision, the article should hence be ready for publication in the International Journal of Molecular Sciences.

general: English use is good but not perfect. Recurring issues include missing articles, use of singular vs. plural, and often overly long and convoluted sentences.

9: Add ‘the’ before ‘second messengers’.

ANSWER: We have changed this as suggested.

27-37: This paragraph should introduce the term ‘particulate’ for the transmembrane ACs. (it actually features as an abbreviation in line 34 but is not explained.)

ANSWER: Thanks for the good suggestion. We have added the ‘particulate’ for the transmembrane GCs in line 39.

figure 1: Could be expanded by popeye proteins which bind cAMP for regulation.

ANSWER: Thanks for the good suggestion. We have added the Popdc in Figure 1 and explained it briefly in line 46-47.

67: What is meant by ‘in subcellular levels’?

ANSWER: We meant to say “in subcellular areas like ER, mitochondria or cytosol microdomains”. We have changed this to “In subcellular microdomains” to avoid misunderstanding.

73: ‘Therapeutic regulation’ is a strange term, consider revising.

ANSWER: We have changed it to ‘Therapeutic targeting of PDEs’

84-86: Unclear why this inhibitor is described in that much molecular detail when nowhere else in the manuscript this would be the case.

ANSWER: We have deleted this part to make the text more clear and coherent.

100-108: Not sure whether this paragraph should not be moved to the subsequent chapter.

ANSWER: We have moved to the beginning of the subsequent chapter. Lines 123-131.

122: Rephrase ‘... leaves the G_alpha protein in its GDP-bound form, which ...’.

ANSWER: This was changed as suggested.

157-159: Detail on original construction of LAPD not needed to this extent, i.e. remove this sentence.

ANSWER: Here we removed this sentence as suggested.

175-178: This part should be moved to the beginning of the section.

ANSWER: We have moved this to the fore part. Lines 178-182. Note that the bigger line No. here is because of new contents were added in the fore part.

211: ‘abundantly exist’ is an exaggeration. As requested above, a brief list of available PAC/PGC options would help.

ANSWER: We removed the “abundantly” and added a new brief section 4 of “Light-regulated nucleotide cyclases” (line 235-257).

214: ‘engineering of new’

ANSWER: This was changed as suggested.

257: ‘suite’

ANSWER: This was changed as suggested.

262: ‘translocation’

ANSWER: This was changed as suggested.

339-344: This part would better fit in chapter 4.

ANSWER: We moved this part to the end of chapter 4.2 to make it more coherent.

325-338: Two recent papers by Wachten (pubmed ids 32579112 and 31252584) seem highly relevant for subcellular localization and issues connected to targeting of optogenetic tools.

ANSWER: The two recent papers are indeed closely related to our manuscript and we have added them in lines 386-390.

In the end, we thank you again for all the constructive reviews.
